# A Blockchain-Applied Personal Health Record Application: Development and User Experience

**Ji Woong Kim** [1], **Su Jin Kim** [2], **Won Chul Cha** [1,3,4] and **Taerim Kim** [3,*]

1   Department of Digital Health, Samsung Advanced Institute for Health Science & Technology, Sungkyunkwan University, Seoul 06355, Korea; rlawlwoong@g.skku.edu (J.W.K.); wc.cha@samsung.com (W.C.C.)
2   Smart Health Lab, Research Institute of Future Medicine, Samsung Medical Center, Seoul 06351, Korea; dh.stellaaa@gmail.com
3   Samsung Medical Center, Department of Emergency Medicine, Sungkyunkwan University School of Medicine, Seoul 06351, Korea
4   Digital Innovation Center, Samsung Medical Center, Seoul 06351, Korea
*   Correspondence: taerimi.kim@samsung.com

**Abstract:** This study aims to introduce a novel blockchain-applied personal health records (PHR) application and validate its user experience. The system transmits the part corresponding to the patient's personal information off-chain and prevents data forgery and falsification by storing encrypted data on-chain. Patients may easily trace the opt-in and opt-out history of their consent data and dynamically store the consent system for data exchange on the blockchain. A mixed-method study using a questionnaire, in-depth interviews, and usability evaluation were conducted for 30 participants. The system usability score was 74.0, indicating the high usability of the application. Those who were familiar with blockchain showed confidence in the application, but those unfamiliar wanted their data to be safe using another way. Most of the participants were interested in exchanging and using their medical data and considered security important but those unfamiliar wanted their data to be safe using another way. We found that participants were concerned about data security and considered a blockchain-based PHR as a novel way to store and exchange their medical information securely. Blockchain is not a visible technology. However, a blockchain-applied PHR must be able to win user trust through visualizations, certificates, and system descriptions.

**Keywords:** medical informatics applications; blockchain; health records; personal

## 1. Introduction

Currently, evidence-based medicine and data-driven analysis are trends in the medical field; thus, the importance of medical data is increasing. Personal health records (PHRs) are important medical data that facilitate better healthcare services and patient safety [1]. Through data, medical staff can efficiently diagnose and treat patients, and reduce unnecessary labs or tests [2]. When regular patients visit an emergency department (ED), medical staff can easily obtain their data. However, in the case of a sudden visit to a nearby hospital, especially in an emergency situation, knowing the patient's history is difficult because patients may not remember their disease and dose histories or laboratory test results [3].

PHR is one of the best ways to gather patient data. In South Korea, most medical records are exchanged using paper or fax. Health information exchange systems are not yet operational because sharing information is limited to a few hospitals and not all data are provided [4–7]. PHR allows patients to self-manage medical information collected through various channels, enabling them to access their medical records anytime and anywhere. In addition, there are advantages, such as the reduction of duplicate tests and prescriptions, improvement of quality of life through personalized health management services, and expansion of medical information exchange between hospitals.

Security and reliability are important factors in medical data and PHRs. A study on PHR recognition by medical staff also showed skepticism about inaccurate data and security issues [8]. There are some possible threats, such as data loss and data manipulation, which could undermine trust in the patient-centered system. Blockchain is an emerging technology to solve these dilemmas, guaranteeing transparency, decentralization, and immutability of the system [9].

Some studies have explored blockchain technology in the healthcare domain. Platforms, such as OmniPHR, MedRec, and FHIRChain are used to transform patient health data sharing in a patient-centric manner [10–12]. Blockchain has also been used to share and store clinical trial data, manage healthcare-related Internet of things, detect medical and research fraud, and conduct public health surveillance [13–15]. In the case of consent data among medical information, data are scattered by institution, and decisions can vary according to each patient's point of view, so a convenient way to safely manage and reflect the dynamic consent system is needed. Although it is an important component in blockchain-applied PHRs, there are few studies related to user experience.

Most studies related to blockchain-applied PHRs have tended to focus on blockchain technology. Castaldo et al. used a multichain platform with electronic hospital record (EHR) data to share health data and improve audit logging securely [16]. Patel et al. used a private blockchain with medical image records to securely share patient medical images, but data search was not considered [17,18]. Fan et al. used a hybrid consensus mechanism based on practical Byzantine fault tolerance but failed to provide sufficient privacy for the patient's identity and energy efficiency [19,20]. Zhu et al. used the Ethereum platform to manage EHR data in a cloud environment, but despite high scalability, it was not practically feasible [21]. Genestier et al. used the Hyperledger platform to manage personal data in e-health, but there was no access control or exhaustive authorization consideration [22–24].

The aim of this study was to introduce a novel blockchain-applied PHR application and to validate its user experience. Thus far, existing research has focused on the technical aspects of blockchain-applied PHR. The novelty of our study is that it not only uses an on-chain, off-chain system to manage patients' consent data in blockchain, but we also researched participants' actual user experience of the blockchain-applied PHR application. We stated blockchain technology to develop a blockchain-applied mobile PHR application. We then conducted a questionnaire to investigate the user experience of the application. Finally, through an in-depth interview, we deeply understood the participants' thoughts on the PHR application regarding blockchain.

## 2. Materials and Methods

### 2.1. Blockchain Technology

Blockchain technology was first described in the form of Bitcoin in a study by Nakamoto [25]. Blockchain is a distributed data storage technology that transparently records transaction details in a ledger and stores them in multiple nodes [26,27]. It is also called a public transaction ledger because all participants in the transaction distribute and disclose the data in a bundle instead of keeping the transaction records only on the central server, as is the existing method in data transactions [28].

Blockchain stores all data in blocks [29]. When a large amount of data is stored in blocks, the speed of the entire network is slowed down, personal information cannot be recorded, entered data cannot be deleted, and participants can view all data, which is a violation of privacy that conflicts with the Personal Information Protection Act [30]. Blockchain solves this problem by using on-chain and off-chain concepts [31].

On-chain refers to all network actions recorded on the blockchain [32]. Off-chain refers to all actions that occur outside the blockchain [33]. As this method does not involve writing directly to the blockchain, it can solve the problem of the blockchain's sloweddown speed or the problem of storing sensitive information, such as personal data [34]. In addition, the data stored off-chain are verified on-chain to ensure the reliability of data forgery prevention [35].

As shown in Figure 1, when a malicious user gains access to the database in a centralized system, data manipulation is possible, and system problems are inevitable [36]. However, in a decentralized system using a blockchain, if data are manipulated by a malicious user, it is easily discernable through data verification with other participants, and the undamaged data can be restored [37]. If a malicious user modifies data by hacking more than half of the nodes in the blockchain network, it can be manipulated, but this is practically impossible [38]. Accordingly, blockchain technology can prevent forgery or falsification of data, thereby enhancing the reliability and stability of the data [39].

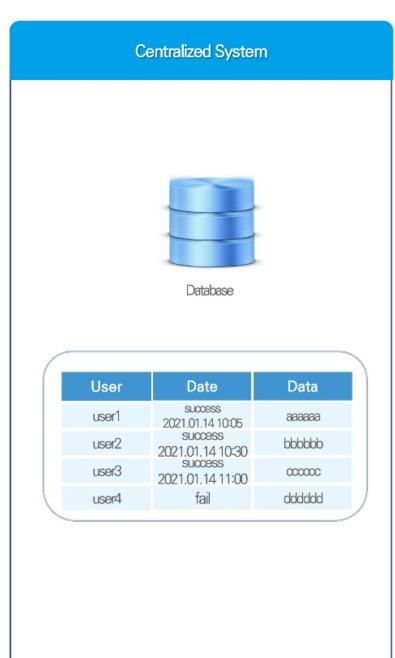
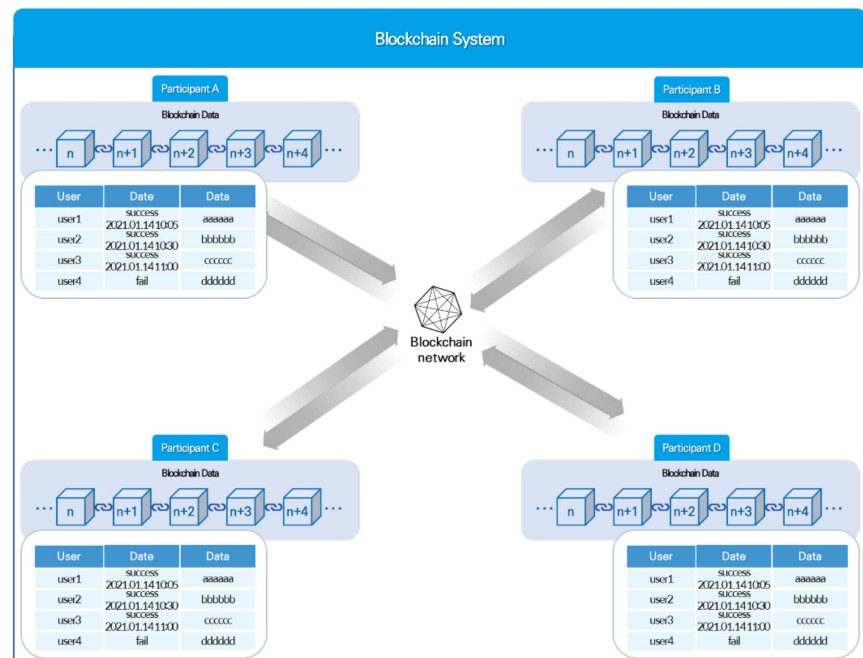

**Figure 1.** Structure of the data storage of the centralized system and blockchain system.

## 2.2. Structure of the Blockchain-Applied PHR

### 2.2.1. Main Function

The blockchain-applied PHR platform has three main roles. First, as shown in Figure 2, the authentication component can confirm the identity of the patient. Patients can connect to the blockchain network using a PHR platform. When signing up on the platform, a personal wallet is requested, and once the personal wallet is issued by the blockchain network, the patient can use it to manage personal data.

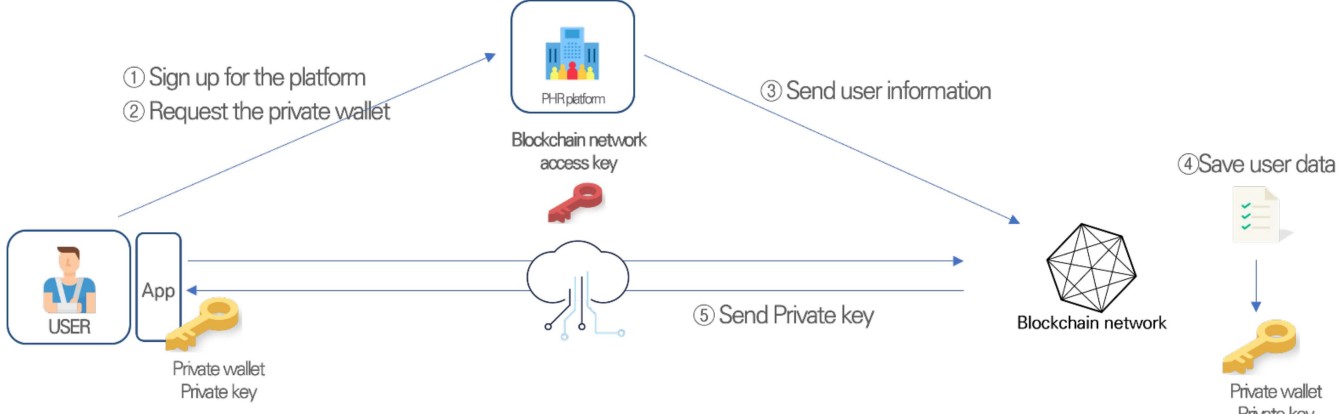

**Figure 2.** System structure to authenticate patient.

Second, as shown in Figure 3, the consent information component checks whether the patient has consented to sharing their medical information. The patient can store consent information using a personal wallet. The consent information is divided into "downloading" and "sharing." By downloading, patients consent to store their medical information. By sharing, patients consent to share health records and medical information with hospitals or other organizations on the platform.

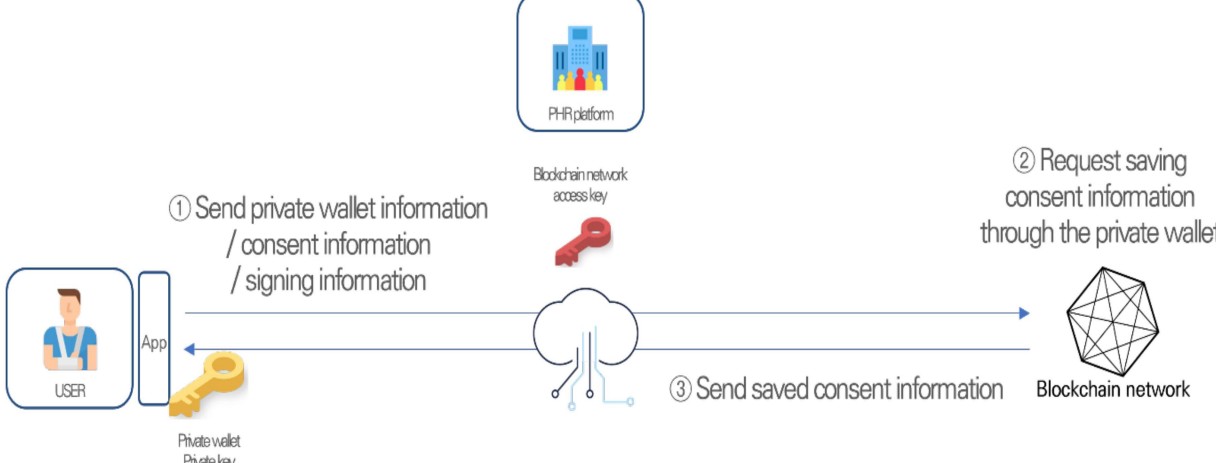

**Figure 3.** System structure of storing consent information in blockchain.

Third, as shown in Figure 4, the data history component verifies who requested medical information and when and checks for data forgery and falsification through the hash value of the transmitted medical information. When a patient requests to download or share medical information, it is recorded on the blockchain network. The provider verifies the patient's consent information on the blockchain, delivers the medical information off-chain, converts the medical information into a hash value, and records it on the blockchain. The platform hashes the medical information received off-chain and compares it to the hash value recorded on the blockchain to check for forgery. When the forgery test is completed, the patient can check their medical information on the platform.

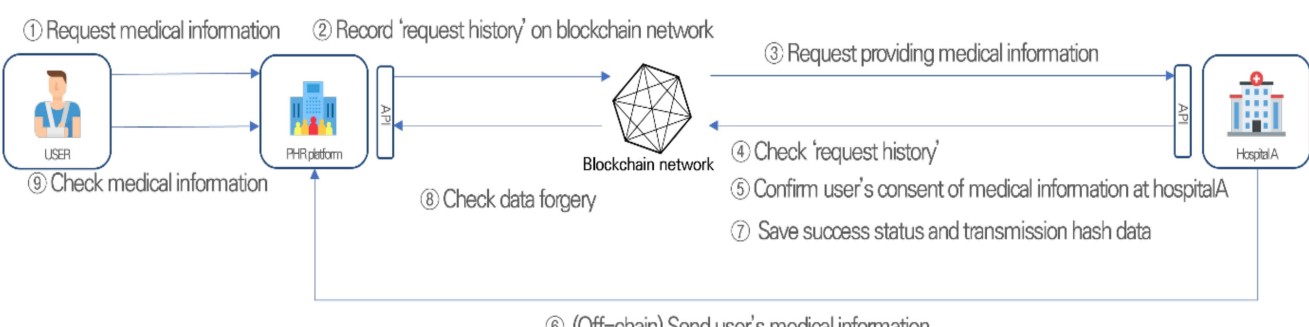

**Figure 4.** System structure of verifying medical information.

### 2.2.2. Data Provided by Blockchain-Applied PHR

Through the application, the patient can check information about the examination and consent form received in the emergency room. The data consist of laboratory examination results, patient consent, and a detailed consent document. The laboratory examination results include visit, treatment, examination, prescription, and discharge data. Patient consent information consists of patient and institution information and consent documents. Detailed information on the consent document consists of basic information on the consent form, information about medical staff, and the content of consent. The patients' laboratory

examination information most frequently referred to in the emergency department was selected. A detailed data item table is shown in Supplementary Table S1.

### 2.2.3. Flow of the Service

After consenting to privacy, the user, who is a patient or caretaker, launches the app and registers with their e-mail and password (Figure 5A).After logging in, the user image, name, blood type, allergies, and most recent ED visits are displayed on the main page. In addition, users can check their medical information from recent emergency department visits, medication to be taken, and health conditions recorded (Figure 5B).

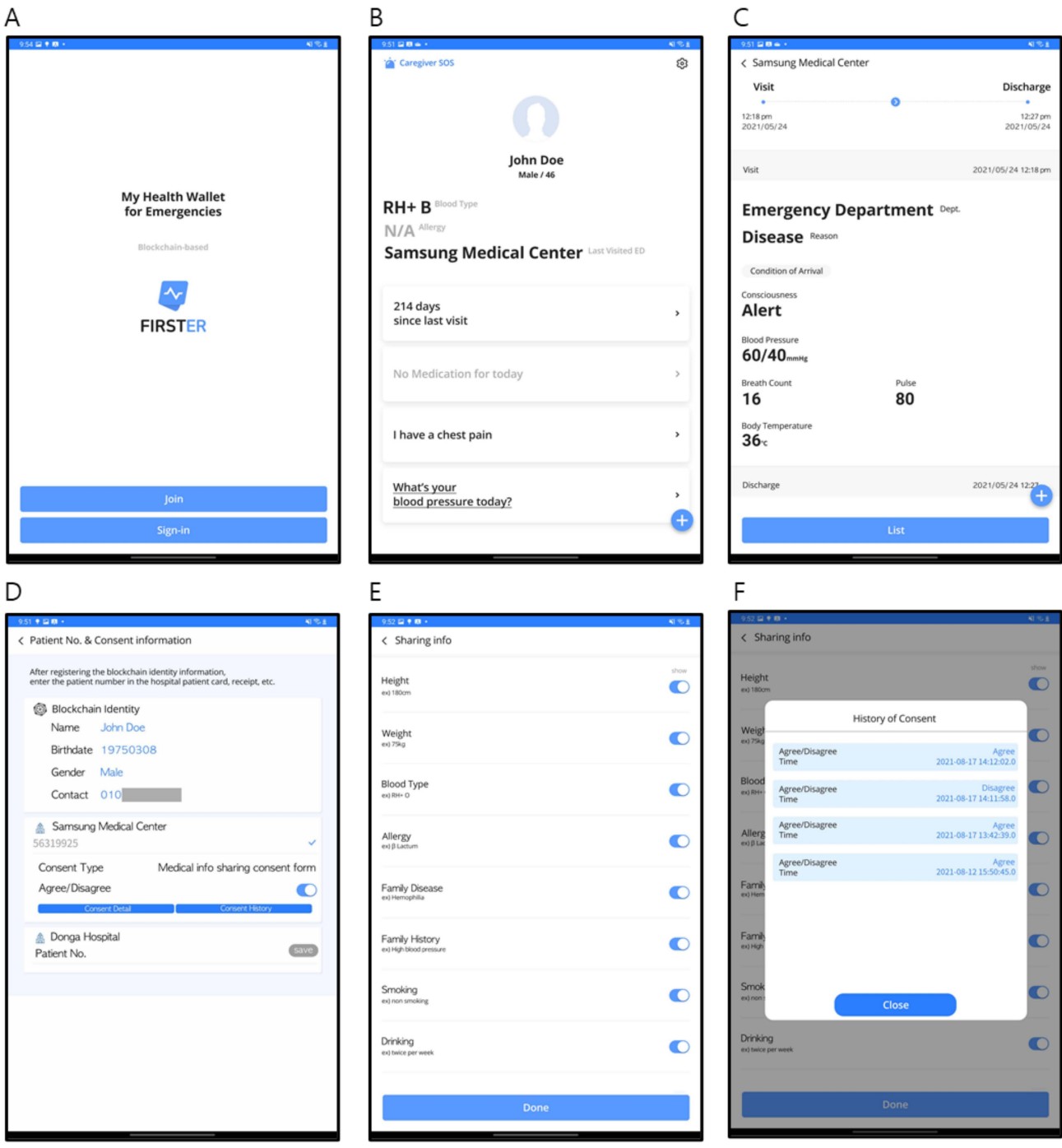

**Figure 5.** Screenshots of the user interface. (**A**) First page of the application for the registration and login; (**B**) main page for checking user information; (**C**) detailed page of patients' medical information,

such as visiting time, test results, and encounter information; (**D**) settings page for managing patients' consent information; (**E**) selection page where the user can choose the items to be shared; (**F**) history page to check whether the user agreed or disagreed to share their medical information.

On the user's medical record page, information, such as the patient's visit data, vital sign data, and lab test results, can be checked along with accurate numbers (Figure 5C).

On the consent settings page, the user can manage their consent information through a blockchain. The patient can check their consent information and details. (Figure 5D).

On the consent selection page, the user can choose the data to be shared. All data lists, such as patient data, encounter data, lab data, prescription data, etc., are shown on the page. Users can approve each item to be shared.

On the history page, the user can check their consent history. The pros and cons of the consent information are stored on the blockchain, and all records can be checked.

### 2.3. Study Design

#### 2.3.1. Participants

The participants in this study worked at a tertiary academic hospital in Seoul, Korea, with 1989 beds. Through convenience sampling, we recruited 30 participants who had medical records at Samsung Medical Center, voluntarily agreed to participate in the study, and were above 19 years of age. Those who did not consent to the study, and those who were judged to have difficulty using mobile applications were excluded. All participants provided informed consent. The study protocol was approved by the Institutional Review Board (IRB) of the Samsung Medical Center in Seoul, Republic of Korea (IRB No. 2021-04-006-001).

#### 2.3.2. Protocol

We conducted user verification with patients who will be subject to the future utilization of this service using a mixed-methods approach. After using the application, the participants completed a paper questionnaire and conducted in-depth interviews. All interviews were recorded with the participants' permission.

#### 2.3.3. Questionnaire Study

The questionnaire consisted of three categories: (1) participants' demographic information, (2) exchange usability evaluation of the application, and (3) awareness of blockchain technology and health information. The awareness and perception of blockchain and dynamic consent categories consisted of 18 questions, including questions about knowledge and familiarity with digital technology, digital literacy, and understanding of the concepts of blockchain and dynamic consent technology. The usability evaluation using platform models utilized the system usability scale (SUS). The SUS target score was above 68 with a total of 10 questions.

#### 2.3.4. In-Depth Interview

Interview questions were about the survey themes, and participant suggestions were requested to gain more knowledge about the subjects. Through an in-depth interview, we evaluated the intention to use, acceptance of the technology, health data management, and suggestions for blockchain-applied PHR applications.

### 2.4. Data Analysis

We determined the demographic features of the study participants. The questions were answered on a Likert scale, and the results were displayed as the mode, as well as numbers and percentages. The SUS assessment tool was calculated according to standardized formulas. Data analyses were performed using the Python software (version 3.7.4).

### 3. Results

#### 3.1. Demographics and SUS Score

A total of 30 patients were enrolled, all of whom answered the questionnaire. Table 1 presents the demographic characteristics of the participants. The same number of female and male patients were included, and most patients were in their twenties and thirties. Most patients visited the hospital for outpatient care.

**Table 1.** Demographic characteristics of participants.

|  |  | N | % |
|---|---|---|---|
| Total |  | 30 | 100 |
| Sex | Male | 15 | 50 |
|  | Female | 15 | 50 |
| Age | 20's | 11 | 36.7 |
|  | 30's | 14 | 46.7 |
|  | >40's | 5 | 16.6 |
| Reason for visit (Duplicate responses are possible) | Outpatient | 30 | 100 |
|  | Emergency | 4 | 13.3 |

We evaluated the user experience of blockchain-applied PHR applications using SUS. The mean SUS score was 74.0, indicating that the application was above average and was good to use. The questionnaire and SUS scores are provided in Supplementary Table S2.

#### 3.2. Questionnaire Study

Most of the participants had heard of blockchain technology, but they did not know how it worked. Participants knew that blockchain technology was good at security, but there were also opinions that blockchain could not be fully trusted because they were not aware of the basic principles of the technology. From the health information exchange perspective, there was a willingness to exchange and utilize their own data, but few people had heard of the dynamic consent system. The participants had a high understanding of health information exchange skills, such as the ability to use digital technology or smartphones. Tables 2 and 3 show the distribution of the questionnaire responses and the modes of the Likert scale scores.

**Table 2.** Distribution, n (%), of blockchain-related survey responses and modes of Likert scale scores (N = 30).

| Category | Question | 1 | 2 | 3 | 4 | 5 | MODE |
|---|---|---|---|---|---|---|---|
| Understanding of blockchain technology | I have heard of blockchain. | 0 | 2 | 0 | 13 | 15 | 5 |
|  | I know how blockchain technology is applied. | 4 | 6 | 9 | 7 | 4 | 3 |
|  | I considered using blockchain to manage my health information. | 5 | 13 | 2 | 5 | 5 | 2 |
| Trust of blockchain technology | I trust blockchain technology. | 1 | 2 | 12 | 10 | 5 | 3 |
|  | I think blockchain technology protects my health information safely. | 0 | 6 | 7 | 13 | 4 | 4 |
|  | I agree that the blockchain transmits data safely. | 2 | 2 | 8 | 14 | 4 | 4 |
|  | I think blockchain technology is more reliable than conventional methods in transmitting health information. | 2 | 1 | 13 | 11 | 3 | 3 |
| Familiarity of blockchain technology | I think blockchain technology has an advantage in accessing my health information. | 1 | 0 | 8 | 11 | 10 | 4 |
|  | I think I am familiar with blockchain technology. | 7 | 13 | 6 | 3 | 1 | 2 |

**Table 3.** Distribution, n (%), of health information exchange-related survey responses and modes of Likert scale scores (N = 30).

| Category | Question | 1 | 2 | 3 | 4 | 5 | MODE |
|---|---|---|---|---|---|---|---|
| Understanding of exchange health information | I have heard of a dynamic consent system. | 22 | 1 | 1 | 4 | 2 | 1 |
| | I was able to actively participate in utilizing and exchanging my health information. | 1 | 1 | 1 | 17 | 10 | 4 |
| | I am interested in accessing my health information independently. | 0 | 0 | 1 | 12 | 17 | 5 |
| Digital literacy | I use the Internet well. | 0 | 1 | 2 | 1 | 26 | 5 |
| | I make good use of digital technology. | 0 | 2 | 2 | 8 | 18 | 5 |
| | I use the applications on my smartphone well. | 0 | 0 | 5 | 6 | 19 | 5 |
| | I can download and install applications well on my phone. | 0 | 1 | 1 | 1 | 27 | 5 |
| | I am active in learning digital technology. | 0 | 0 | 5 | 6 | 19 | 5 |
| | I feel scared when someone talks about digital technology. | 16 | 7 | 5 | 1 | 1 | 1 |

*3.3. In-Depth Interviews*

Using interviews, we learned of the participants' thoughts about blockchain technology and PHR services. We structured our interviews into three key themes: intention of use, acceptance of the technology, and health data management and sharing. Interview protocol is described in Supplementary Table S3. We compared the pros and cons of the user experience for each theme.

3.3.1. Intention of Use

Questions about the intention of use examined the need to use blockchain-applied PHRs continuously. When the participants were asked about their intention to use blockchain and PHR, they offered pros and cons. Participants thought that blockchain was needed to protect the patient's data and for system security. However, there were also opinions that blockchain was not the only way to protect patients' health data.

"I like being able to check my records anywhere without having to go to the hospital." [Study participant 30].

"By using the block chain [sic], the security is strengthened, and it seems to be reliable." [Study participant 24].

"You can check it on the hospital app, too, so I don't know much about the need." [Study participant 10].

"I don't know exactly what blockchain is, so I can't say clearly, but honestly, I think that blockchain technology is really necessary? [sic]." [Study participant 29].

3.3.2. Acceptance of Technology

Questions about the acceptance of technology concerned users' understanding and thoughts about blockchain and PHR. Many participants had only a basic understanding of blockchain or were not familiar with it. Most participants had no difficulty using the PHR, but some needed a manual or intuitive user interface.

"I am only aware of the block chain [sic] technology that is applied to Bitcoin." [Study participant 2].

"I don't have to worry about how to use it because they gave me enough instructions." [Study participant 3].

"It wasn't difficult for me to understand, but I don't know how adults will actually take it." [Study participant 30].

### 3.3.3. Health Data Management and Sharing

For health data management and sharing questions, we asked participants for their opinions about sharing health data using blockchain and PHR.

"When I go to another hospital, I don't have to explain my condition repeatedly, so I think sharing information will be helpful." [Study participant 24].

"By using the blockchain, we feel that data is transmitted securely." [Study participant 14].

### 3.3.4. Suggestions for Improvement

Participants showed a positive response to the blockchain-applied PHR application. They expected the application to improve the quality of treatment and be helpful in communication with medical staff. They offered the following suggestions: (1) all hospitals—primary, secondary, and tertiary—should be linked; (2) user convenience will increase if there is an insurance claim function; and (3) for those who do not have a good understanding of blockchain, a certification of the application's security could make it more trustworthy.

"If primary hospital, secondary hospital, tertiary hospital is linked, I am willing to use it." [Study participant 10].

"If it is certified that there is no problem in security that blockchain technology has been applied through a government or institution, it is of course trustworthy." [Study participant 28].

### 3.3.5. Summary of the In-Depth Interviews

A summary of the in-depth interview is presented in Table 4.

**Table 4.** Summary of the pros and cons of blockchain technology and PHR service as reported by users.

| Key Theme | Category | Pros | Cons |
|---|---|---|---|
| Intention of use | Blockchain | Blockchain is needed to protect patient data. Blockchain is needed because it is good in terms of system security. | Wondering if blockchain is the only way to protect patients' information. |
| | PHR | Exchanging patients' health information is needed. Checking my health data anywhere and anytime is needed. | App provided by the hospital can also be used to check patient health data. |
| Acceptance of technology | Blockchain | A basic understanding of blockchain technology. Users can have sovereignty over their own data. | Does not know about the technology. Heard of the technology but does not know about the mechanism. |
| | PHR | No difficulty in using service. | A manual on how to use the service is needed. Application design needs to be a little more intuitive. |
| Health data management and sharing | Blockchain | Health data could be systemically managed. Able to independently manage my own health data. | Negative opinion due to not knowing much about the technology. |
| | PHR | Sharing information with medical staff has become easy. | Not very interested. |

## 4. Discussion

In this study, we introduced a blockchain-applied PHR application that enables the exchange of data from hospitals and empowers patients to manage their own data. Through this application, patients can download their data from hospitals to their own application

and can provide reliable information to medical staff anytime and anywhere. Although the application needs more study and validation, it shows that medical information can be stored separately on-chain and off-chain, and thus data security and system stability can be maintained. In addition, we were able to confirm the user experience of the blockchain and PHR through interviews.

Unnecessary medical care, such as redundant laboratory testing, procedures, and opioid overuse, is often caused by not sharing patient health records, eventually putting patients' safety at risk [40,41]. A blockchain-applied PHR is one of the most effective and safe ways to collect patient data. Participants could access their information at any time and from any location using a blockchain-applied PHR, which allows them to self-manage medical data collected through different channels. Additional advantages might include higher healthcare quality through tailored health management services avoiding duplication of medical care, which enables medical information exchange between institutions.

Blockchain technology has been used to exchange medical data in previous studies. A study showed the exchange and the management of medical data using a Hyperledger platform, but the privacy was unsecured [26–28]. Another study used the Ethereum platform to manage EHR data in a cloud setting; however, it was not feasible [25]. Few studies evaluated the user experience of PHR applications using blockchain technology among these studies. This study did not just develop a system but conducted the user experience study with a mixed-method study to listen to the actual voice of users. As a result, authors could easily exchange their medical data through a blockchain-applied PHR application with the confidence of sovereignty on their data.

On-chain and off-chain data management are important when applying blockchain to medical data [11]. Among medical data, consent-related data should be easy to opt in and out and should be carefully managed [42]. The patient should be able to arbitrarily change consent information, and the change should be safely managed through the blockchain [43]. As it is difficult to delete content that has been uploaded to the blockchain, the blockchain can prevent forgery and falsification by encrypting the patient's personal information or medical information that can identify the patient [44]. By managing patient data on-chain and off-chain, it is possible to maintain the data security, speed, and stability of the system [45]. Our blockchain-applied PHR application manages medical data using on-chain and off-chain, ensuring user privacy and a consistent data transfer speed.

This study shows the user experience pros and cons of the blockchain-applied PHR. Most participants knew blockchain technology through the media or Bitcoin. Although they did not know the details about blockchain technology, there were many opinions that it would be safe. Participants with knowledge of blockchain technology said that it would be helpful in preventing data forgery and maintaining system security. In terms of PHR, participants felt the need to check their data, regardless of place or time. They said that it was a necessary service for the elderly and that it would be of great help in monitoring their condition. Meanwhile, some argued that blockchain was not the only solution to the security problem or that services made by trusted organizations would be more reliable. They also stated that they were not interested in checking their health data or that they could check their information in the app provided by the hospital.

We also identified the challenges in the application. There was a question regarding whether the data were stored safely because blockchain technology is invisible. Using blockchain technology, data forgery can be checked, so patient data can be safely protected, but it is difficult for users to do so [46]. Therefore, if a trusted institution confirms the security of the system or certifies the application, it would be trustworthy.

## 5. Limitations and Future Work

This study has several limitations. First, it was conducted over a short period during the coronavirus-19 pandemic. Therefore, the study population may be different from that in a non-pandemic situation. Second, it was a single-center trial; therefore, selection bias could not be excluded. External validation is needed when applying the findings in other settings,

even though wide variations of participants could be recruited from an academic tertiary referral hospital with 1,989 beds in Seoul, Korea. Third, the number of recruited participants was limited to express the voice of the users. Based on Denzin and Lincoln, the number of subjects expected to reach saturation in existing qualitative studies is usually suggested to be approximately 10 to 20 people [47]. Thus, we believe that the number of interview participants was sufficient. Fourth, we did not observe the satisfaction or feasibility of each function of the application. However, what we examined was the effect of the blockchain-applied PHR on users in the context of a newly adopted technology. A structured in-depth interview with three key themes, intention of use, acceptance of technology, and health data management and sharing, was conducted to evaluate this perspective.

In future studies, in order to supplement the above-mentioned limitations, we will recruit more diverse participants from multiple institutions. In addition, we plan to evaluate the user experience of the blockchain-applied PHR through a long-term period research. We plan to investigate the detailed user experience of each function of the application by analyzing the log data of the participants, and through this, we will be able to find the usability and necessity of each function.

## 6. Conclusions

This study introduced a blockchain-applied PHR application that allows patients to manage their medical information using blockchain technology. Using blockchain, we developed a secure system for patients to exchange medical data. In particular, it was possible to store and manage medical data effectively on and off chains. Through a mixed-method study, we found that participants were concerned about data security and considered a blockchain-applied PHR as a novel way to store and exchange their medical information securely. Blockchain is not a visible technology. However, to have come a long way for the secure exchange of medical information, a blockchain-applied PHR must be able to gain users' trust through visualizations, certificates, and system descriptions.

**Supplementary Materials:** The following are available online at https://www.mdpi.com/article/10.3390/app12041847/s1, Table S1: A detailed data item list provided by Blockchain-Applied PHR, Table S2: The questionnaire and SUS scores of all participants, Table S3: Semi-structed interview protocol of the study.

**Author Contributions:** Conceptualization, J.W.K.; writing the manuscript, S.J.K.; interviews, W.C.C. and T.K.; supervising the research work. All authors have read and agreed to the published version of the manuscript.

**Funding:** This paper was written based on the contents of the research project (HI19C0552) "Channel-type personal health record platform based on blockchain" of the Korea Health Industry Development Institute (KHIDI).

**Institutional Review Board Statement:** The study protocol was approved by the Institutional Review Board (IRB) of the Samsung Medical Center in Seoul, Republic of Korea (IRB No.2021-04-006-001).

**Informed Consent Statement:** Written informed consent has been obtained from the patient(s) to publish this paper.

**Acknowledgments:** We would like to thank V.T.W. for supporting the development of blockchain-applied PHRs.

**Conflicts of Interest:** The authors declare no conflict of interest.

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
