# Peer review of "A Blockchain-Applied Personal Health Record Application: Development and User Experience"

_applsci, doi:10.3390/app12041847_

Round 1

Reviewer 1 Report

The manuscript entitled "Developing a Blockchain-Applied Personal Health Record Application: Development and Usability Study" represents a study in which a blockchain-based personal health record (PHR) application is developed to allow patients to securely manage their medical information using blockchain technology. Moreover, the usability of the application was tested on a group of 30 hospital patients and evaluation results are reported.

The manuscript is well written and easy to follow. The proposed method is technically sound and appropriately described. The proposed method has a straightforward, practical application in the medical centers for patient data security and exchange. Moreover, the proposed method is validated based on the data acquired from patients participating in its testing. The obtained results are clearly presented and appropriately discussed, indicating the proposed method's practical applicability. The obtained results support the conclusions and future research directions are also provided.

Here are some comments I would like the authors to address before the manuscript is considered for publication:

  1. In its current form, the paper resembles more a technical study and product introduction than an original research article. Please, extend the Introduction section by a more detailed literature review of the recent studies in this specific field. Please also better emphasize the originality and novelty of the proposed approach in context of the existing research.
  2. Please provide in the Introduction section a paragraph describing the structure of the paper.

Author Response

The authors have revised English grammar, with awkward expressions and grammatical errors throughout the manuscript. Please note that the correction has been reflected by turning on the tracking function. We changed a title more concisely from ‘Developing a blockchain-applied personal health record application: development and usability study’ to ‘A Blockchain-Applied Personal Health Record Application: Development and User Experience’

Point 1: In its current form, the paper resembles more a technical study and product introduction than an original research article. Please, extend the Introduction section by a more detailed literature review of the recent studies in this specific field. Please also better emphasize the originality and novelty of the proposed approach in context of the existing research.

 Response 1: Thank you for your valuable comment. As you commented we extended introduction section by a more literature review. Most of the blockchain-related PHR studies have been focused on the technical part. We added a paragraph about the charateristics of blockchain applied PHR by other papers in the introduction section through literature review. Also, in the next paragraph, we emphasized the originality and novelty of our research. The aim of this study was to introduce a novel blockchain-applied PHR application, and to validate the user experience of that. The existing research so far has focused on the technical aspects of blockchain-applied PHR, the novelty of our study is that not only used on-chain, off-chain system and managing patient’s consent data in blockchain, but also researched the participants actual usability experience of the blockchain-applied PHR application.

Point 2: Please provide in the Introduction section a paragraph describing the structure of the paper.

Response 2: We apologize for the inadequate explanation. We tried adding the structure of our paper to the introduction section step by step. First, We described blockchain technology to develop a blockchain-applied mobile PHR application. And we conducted a questionnaire to investigate the usability of the application among them. Finally, through an in-depth interview, we deeply understood the participants' thoughts on the PHR application which applied the blockchain

Reviewer 2 Report

The paper presents a hot area of research and has a number of strengths but needs the following improvements to be addressed.

  1. Does the 20 participants not a small dataset for validating the proposed approach?
  2. The contributions of the study are missing. Need to be added at appropriate places.
  3. The organization paragraph of the study is missing in the introduction section. Add that.
  4. The questionar is user consent based, so if a participants agree with a sub section of the questionnaire and regret then what should be the criteria for implementation of the study?
  5. Summary results should be added in the abstract.
  6. Limitations of the existing studies are missing and need to be highlighted in literature review section.
  7. Future work should be highlighted to let the readers know about the possible future research.
  8. English grammar shall be revisited.

Author Response

The authors have revised English grammar, with awkward expressions and grammatical errors throughout the manuscript. Please note that the correction has been reflected by turning on the tracking function. We changed a title more concisely from ‘Developing a blockchain-applied personal health record application: development and usability study’ to ‘A Blockchain-Applied Personal Health Record Application: Development and User Experience’

Point 1: Does the 20 participants not a small dataset for validating the proposed approach?

Response 1: Thank you for your valuable comment. We recruited 30 participants in a tertiary academic hospital in Seoul, Korea, with 1989 beds. Based on Denzin and Lincoln, the number of subjects expected to be saturated in existing qualitative studies is usually suggested to be about 10 to 20 people [1]. We believe the number of participants of 30 for the interview were proper. However, since this is also a limitation of our study, in the future studies we will recruit more participants through a multicenter study.

  1. Denzin, Norman K., Yvonna S. Lincoln, and Michael D. Giardina. "Disciplining qualitative research." International journal of qualitative studies in education 19.6 (2006): 769-782.

Point 2: The contributions of the study are missing. Need to be added at appropriate places.

Response 2: Thank you for your comment. We described author contributions below the conclusions section. “Author Contributions: Conceptualization, J.W.K; writing the manuscript, S.J.K; performing inter-view, W.C.C. and T.K.; supervising the research work. All authors have read and agreed to the published version of the manuscript.”

Point 3: The organization paragraph of the study is missing in the introduction section. Add that.

Response 3: We apologize for the inadequate explanation. We tried adding the structure of our paper to the introduction section step by step. In the last pharagraph of the introduction section we added,

“We described blockchain technology to develop a blockchain-applied mobile PHR application. And we conducted a questionnaire to investigate the usability of the application among them. Finally, through an in-depth interview, we deeply understood the participants' thoughts on the PHR application which applied the blockchain”

Point 4: The questionar is user consent based, so if a participants agree with a sub section of the questionnaire and regret then what should be the criteria for implementation of the study?

Response 4: Thank you for your comment. Fortunately, in our study, all participants participated in the questionnaire without regret. However, if among the participants, if their consent was changed in the middle, we would have excluded the participant's score, and we would have collected the participant's opinion through an interview.

Point 5: Summary results should be added in the abstract.

Response 5: Thank you for your valuable comment. We editted abstract section including summary results.

“This study aims to introduce a novel blockchain-applied personal health records (PHR) application and evaluate its usability. The system transmits the part corresponding to the patient's personal information off-chain and prevents data forgery and falsification by storing encrypted data on-chain. Patients may easily trace the opt-in and opt-out history of their consent data and dynamically store the consent system for data exchange on the blockchain. A mixed-method study using a questionnaire, in-depth interviews, and usability evaluation were conducted for 30 participants. The system usability score was 74.0, indicating the high usability of the application. Those who were familiar with blockchain showed confidence in the application, but those unfamiliar wanted their data to be safe using another way. Most of the participants were interested in exchanging and using their medical data and considered security important but those unfamiliar wanted their data to be safe using another way. We found that participants were concerned about data security and considered a blockchain-based PHR as a novel way to store and exchange their medical information securely. Blockchain is not a visible technology. However, a blockchain-applied PHR need be able to win user trust through visualizations, certificates, and system descriptions.”

Point 6: Limitations of the existing studies are missing and need to be highlighted in literature review section.

Response 6: Thank you for your valuable comment. We added limitation and future work section below the discussion section.

“This study has several limitations. First, it was conducted over a short period during the coronavirus-19 pandemic. Therefore, the study population may be different from that in a non-pandemic situation. Second, it was a single-center trial; therefore, selection bias could not be excluded. External validation is needed when applying the findings in other settings, even though wide variations of participants could be recruited from an academic tertiary referral hospital with 1,989 beds in Seoul, Korea. Third, the number of recruited participants was limited to express the voice of the users. Based on Denzin and Lincoln, the number of subjects expected to reach saturation in existing qualitative studies is usually suggested to be approximately 10 to 20 people [48]. Thus, we believe that the number of interview participants was sufficient. Fourth, we did not observe the satisfaction or feasibility of each function of the application. However, what we examined was the effect of the blockchain-applied PHR on users in the context of a newly adopted technology. A structured in-depth interview with three key themes – intention of use, acceptance of technology, and health data management and sharing – was conducted to evaluate this perspective.”

Point 7: Future work should be highlighted to let the readers know about the possible future research.

Response 7: Thank you for your valuable comment. We added limitation and future work section below the discussion section.

“In the future study, in order to supplement the above-mentioned limitations, we will recruit more diverse participants from multiple institutions. In addition, we plan to evaluate the usability of the blockchain-applied PHR through a long-term period re-search. We plan to investigate the detailed usability of each function of the application by analyzing the log data of the participants, and through this, we will be able to find the usability and necessity of each function.”

Point 8: English grammar shall be revisited.

Response 8: Thank you for your comment. Currently, we did grammar correction through a company called 'Editage'.

Reviewer 3 Report

This study introduced a blockchain-applied PHR application that allows patients to manage their medical information using blockchain technology. Using blockchain, authors developed a secure system for patients to exchange their medical data. 

It is an interesting paper on emerging theme in health information system area. However it should be improved on following issues.

Motivation of this study supported by related references should be included in the Introduction section. Also structure of the paper should be in the end of Introduction section.

Research method selection approach should be described.

Interview protocol should be described preferably in tabular form.

Related work section is very weak. Many recent related references have not been included. There are only very  limited of references and very few from reputed journals. Therefore, authors should include these in section 2.  Due to this Discussion section is also weak (No comparison/map with related work)

Discussion section is weak as it presents mere observations of this study. There are no inputs from related studies to know this study results map with similar or near similar studies. Whether these are similar, divergent or mixed ones. Also authors should include their insight too. What were main challenges during interview process.

Conclusion section should include Limitations and future research  directions for readers. Also novelty of the paper should be outline in both Abstract and Conclusion section.  

In general English is good. However revised version should be carefully edited by native language/professional proof reader. See for instance:  Data analyses were performed using the Python

Author Response

The authors have revised English grammar, with awkward expressions and grammatical errors throughout the manuscript. Please note that the correction has been reflected by turning on the tracking function. We changed a title more concisely from ‘Developing a blockchain-applied personal health record application: development and usability study’ to ‘A Blockchain-Applied Personal Health Record Application: Development and User Experience’

Point 1: Motivation of this study supported by related references should be included in the Introduction section. Also structure of the paper should be in the end of Introduction section.

Response 1: Thank you for your valuable comment. As you commented we extended introduction section by a more literature review. Also, we tried adding the structure of our paper to the introduction section step by step. First, We described blockchain technology to develop a blockchain-applied mobile PHR application. And we conducted a questionnaire to investigate the usability of the application among them. Finally, through an in-depth interview, we deeply understood the participants' thoughts on the PHR application which applied the blockchain

Point 2: Research method selection approach should be described.

Response 2: Thank you for your comment. The authors addressed the mixed-method study as a research method selection approach. We revised abstract and the following paragraph describing the study protocol for those unfamiliar with the mixed-method study.

  • “This study aims to introduce a novel blockchain-applied personal health records (PHR) application and evaluate its usability. The system transmits the part corresponding to the patient's personal information off-chain and prevents data forgery and falsification by storing encrypted data on-chain. Patients may easily trace the opt-in and opt-out history of their consent data and dynamically store the consent system for data exchange on the blockchain. A mixed-method study using a questionnaire, in-depth interviews, and usability evaluation were conducted for 30 participants. The system usability score was 74.0, indicating the high usability of the application. Those who were familiar with blockchain showed confidence in the application, but those unfamiliar wanted their data to be safe using another way. Most of the participants were interested in exchanging and using their medical data and considered security important but those unfamiliar wanted their data to be safe using another way. We found that participants were concerned about data security and considered a blockchain-based PHR as a novel way to store and exchange their medical information securely. Blockchain is not a visible technology. However, a blockchain-applied PHR need be able to win user trust through visualizations, certificates, and system descriptions.”
  • 3.2 Protocol

We conducted user verification with patients who will be subject to the future utilization of this service using a mixed-methods approach. After using the application, the participants completed a paper questionnaire for quantitative data and conducted in-depth interviews for qualitative data. All interviews were recorded with the participants' permission.

Point 3: Interview protocol should be described preferably in tabular form.

Response 3: Thank you for your valuable comment. We added our interview protocol in tabular form at the supplementary table1

Key theme

Category

Question

Intention of use

Blockchain

Do you feel the need for blockchain technology?

What are the security issues caused by blockchain technology?

PHR

Do you feel the need for a patient-centric PHR platform?

Do you willing to use PHR services?

Acceptance of technology

Blockchain

Do you know how blockchain technology works?

What are the benefits of using blockchain technology?

What are the concerns with blockchain technology?

PHR

Is the information provided by the app clear?

Do you understand the information provided by the app?

Do you feel convenient for finding features within the app?

Health data management and sharing

Blockchain

If you could utilize data using blockchain technology, what would it be?

Do you think it is safe to use blockchain technology to make data available?

PHR

Are you interested in using your personal health information independently?

What are the important factors in sharing personal health information?

Table 1 Semi-structed interview protocol of the study.

Point 4: Related work section is very weak. Many recent related references have not been included.

Response 4: Thank you for your comment. As you commented we extended introduction section by a more literature review. Most of the blockchain-related PHR studies have been focused on the technical part. We added a paragraph about the charateristics of blockchain applied PHR by other papers in the introduction section through literature review. Also, in the next paragraph, we emphasized the originality and novelty of our research. The aim of this study was to introduce a novel blockchain-applied PHR application, and to validate the user experience of that. The existing research so far has focused on the technical aspects of blockchain-applied PHR, the novelty of our study is that not only used on-chain, off-chain system and managing patient’s consent data in blockchain, but also researched the participants actual usability experience of the blockchain-applied PHR application.

Point 5: Discussion section is weak as it presents mere observations of this study. There are no inputs from related studies to know this study results map with similar or near similar studies. Whether these are similar, divergent or mixed ones. Also authors should include their insight too. What were main challenges during interview process.

Response 5: Thank you for your valuable comment. We supplemented our discussion section. We have added the contents of related research to the discussion section. Also, one of the biggest challenges was to explain invisible techniques to interviewees, and the measures for this were additionally described.

"In this study, we introduced a blockchain-applied PHR application, that enables the exchange of data from hospitals and empowers patients to manage their own data. Through this application, patients can download their data from hospitals to their own application and can provide reliable information to medical staff anytime and anywhere. Although the application needs more study and validation, it shows that medical information can be stored separately on-chain and off-chain, and thus data security and system stability can be maintained. In addition, we were able to confirm the user experience of the blockchain and PHR through interviews.

Unnecessary medical care, such as redundant laboratory testing, procedures, and opioid overuse, is often caused by not sharing patient health records, eventually putting patients' safety at risk [40,41]. A blockchain-applied PHR is one of the most effective and safe ways to collect patient data. Participants could access their information at any time and from any location using a blockchain-applied PHR, which allows them to self-manage medical data collected through different channels. Additional advantages might be included higher healthcare quality through tailored health management ser-vices avoiding duplication of medical care, which enables medical information exchange between institutions.

Blockchain technology has been used to exchange medical data in previous studies. A study showed the exchange and the management of medical data using a Hyperledger platform, but the privacy was unsecured [26-28]. Another study used the Ethereum platform to manage EHR data in a cloud setting; however, it was not feasible [25]. Few studies evaluated the usability of PHR applications using blockchain technology among these studies. This study did not just develop a system but conducted the usability study with a mixed-method study to listen to the actual voice of users. As a result, authors could find participants easily exchanges their medical data through a blockchain-applied PHR application with the confidence of sovereignty on their data.

On-chain and off-chain data management are important when applying blockchain to medical data [42]. Among medical data, consent-related data should be easy to opt-in and out and should be carefully managed [43]. The patient should be able to arbitrarily change consent information and the change should be safely managed through the blockchain [44]. As it is difficult to delete content that has been uploaded to the blockchain, the blockchain can prevent forgery and falsification by encrypting the patient's personal information or medical information that can identify the patient [45]. By managing patient data on-chain and off-chain, it is possible to maintain the data security, speed, and stability of the system [46]. Our blockchain-applied PHR application manages medical data using on-chain and off-chain, ensuring user privacy and a consistent data transfer speed.

This study shows the user experience pros and cons of the blockchain-applied PHR. Most participants knew blockchain technology through the media or Bitcoin. Although they did not know the details about blockchain technology, there were many opinions that it would be safe. Participants with knowledge of blockchain technology said that it would be helpful in preventing data forgery and maintaining system security. In terms of PHR, participants felt the need to check their data, regardless of place or time. They said that it was a necessary service for the elderly and that it would be of great help in monitoring their condition. Meanwhile, some argued that blockchain was not the only solution to the security problem or that services made by trusted organizations would be more reliable. They also stated that they were not interested in checking their health data or that they could check their information in the app provided by the hospital.

We also identified the challenges in the application. There was a question regarding whether the data were stored safely because blockchain technology is invisible. Using blockchain technology, data forgery can be checked, so patient data can be safely protected, but it is difficult for users to do so [47]. Therefore, if a trusted institution confirms the security of the system or certifies the application, it would be trustworthy."

Point 6: Conclusion section should include Limitations and future research  directions for readers. Also novelty of the paper should be outline in both Abstract and Conclusion section

Response 6: Thank you for your comment. We added limitation and future work section below the discussion section. Also, we edited our abstract to show our study’s novelty.

Limitations and Future Work

This study has several limitations. First, it was conducted over a short period during the coronavirus-19 pandemic. Therefore, the study population may be different from that in a non-pandemic situation. Second, it was a single-center trial; therefore, selection bias could not be excluded. External validation is needed when applying the findings in other settings, even though wide variations of participants could be recruited from an academic tertiary referral hospital with 1,989 beds in Seoul, Korea. Third, the number of recruited participants was limited to express the voice of the users. Based on Denzin and Lincoln, the number of subjects expected to reach saturation in existing qualitative studies is usually suggested to be approximately 10 to 20 people [48]. Thus, we believe that the number of interview participants was sufficient. Fourth, we did not observe the satisfaction or feasibility of each function of the application. However, what we examined was the effect of the blockchain-applied PHR on users in the context of a newly adopted technology. A structured in-depth interview with three key themes – intention of use, acceptance of technology, and health data management and sharing – was conducted to evaluate this perspective.

In the future study, in order to supplement the above-mentioned limitations, we will recruit more diverse participants from multiple institutions. In addition, we plan to evaluate the usability of the blockchain-applied PHR through a long-term period research. We plan to investigate the detailed usability of each function of the application by analyzing the log data of the participants, and through this, we will be able to find the usability and necessity of each function.

Abstract

This study aims to introduce a novel blockchain-applied personal health records (PHR) application and validate its user experience. The system transmits the part corresponding to the patient's personal information off-chain and prevents data forgery and falsification by storing encrypted data on-chain. Patients may easily trace the opt-in and opt-out history of their consent data and dynamically store the consent system for data exchange on the blockchain. A mixed-method study using a questionnaire, in-depth interviews, and usability evaluation were conducted for 30 participants. The system usability score was 74.0, indicating the high usability of the application. Those who were familiar with blockchain showed confidence in the application, but those unfamiliar wanted their data to be safe using another way. Most of the participants were interested in exchanging and using their medical data and considered security important but those unfamiliar wanted their data to be safe using another way. We found that participants were concerned about data security and considered a blockchain-based PHR as a novel way to store and exchange their medical information securely. Blockchain is not a visible technology. However, a blockchain-applied PHR need be able to win user trust through visualizations, certificates, and system descriptions.

Round 2

Reviewer 2 Report

Point-4 is not properly addressed. Contributions of the study is not provided. Authors' contributions added after conclusion is not what I meant. You have to clrealy provide what are the uniqueness and contributions you made in this study.

Author Response

Thank you so much for taking the time to read our study carefully. The review made our paper better.

Response : We apologize for the misunderstanding. Research related to the existing blockchain-applied PHR has been focused on the blockchain technology applied to PHR. Our study's uniqueness and contribution is that through the mix-method study, we investigated not only the technical part of blockchain applied PHR but also the user experience. Following paragraphs and sentences are added in the discussion section.

Unnecessary medical care, such as redundant laboratory testing, procedures, and opioid overuse, is often caused by not sharing patient health records, eventually putting patients' safety at risk [40,41]. A blockchain-applied PHR is one of the most effective and safe ways to collect patient data. Participants could access their information at any time and from any location using a blockchain-applied PHR, which allows them to self manage medical data collected through different channels. Additional advantages might be included higher healthcare quality through tailored health management ser-vices avoiding duplication of medical care, which enables medical information exchange between institutions.
Blockchain technology has been used to exchange medical data in previous studies. A study showed the exchange and the management of medical data using a Hy-perledger platform, but the privacy was unsecured [26-28]. Another study used the Ethereum platform to manage EHR data in a cloud setting; however, it was not feasible [25]. Few studies evaluated the usability of PHR applications using blockchain technology among these studies. This study did not just develop a system but conducted the usability study with a mixed-method study to listen to the actual voice of users. As a result, authors could find participants easily exchanges their medical data through a blockchain-applied PHR application with the confidence of sovereignty on their data.

Also, we supplemented our conclusion section.

This study introduced a blockchain-applied PHR application that allows patients to manage their medical information using blockchain technology. Using blockchain, we developed a secure system for patients to exchange medical data. In particular, it was possible to store and manage medical data effectively on and off chains. Through a mixed-method study, we found that participants were concerned about data security and considered a blockchain-applied PHR as a novel way to store and exchange their medical information securely. Blockchain is not a visible technology. However, to have come a long way for the secure exchange of medical information, a blockchain-applied PHR must be able to gain users’ trust through visualizations, certificates, and system descriptions.

Reviewer 3 Report

Authors have addressed mentioned comments

Author Response

Thank you so much for taking the time to read our study carefully. The review made our paper better.